# MicroRNA-20a-5p Downregulation by Atorvastatin: A Potential Mechanism Involved in Lipid-Lowering Therapy

**DOI:** 10.3390/ijms23095022

**Published:** 2022-04-30

**Authors:** Kathleen Saavedra, Karla Leal, Nicolás Saavedra, Yalena Prado, Isis Paez, Carmen G. Ubilla, Gabriel Rojas, Luis A. Salazar

**Affiliations:** 1Center of Molecular Biology and Pharmacogenetics, Department of Basic Sciences, Faculty of Medicine, Universidad de La Frontera, Temuco 4811230, Chile; kathleen.saavedra@ufrontera.cl (K.S.); k.leal.villegas@gmail.com (K.L.); nicolas.saavedra@ufrontera.cl (N.S.); yalenapradovizcaino@gmail.com (Y.P.); isis.paez921007@gmail.com (I.P.); carubilla@yahoo.com (C.G.U.); g.alfonso.r@gmail.com (G.R.); 2Scientific and Technological Bioresource Nucleus (BIOREN), Universidad de La Frontera, Temuco 4811230, Chile

**Keywords:** microRNA, statin, hypercholesterolemia, cardiovascular diseases, atorvastatin

## Abstract

The treatment of hypercholesterolemia is mainly based on statins. However, the response to pharmacological therapy shows high inter-individual variability, resulting in variable effects in both lipid lowering and risk reduction. Thus, a better understanding of the lipid-lowering mechanisms and response variability at the molecular level is required. Previously, we demonstrated a deregulation of the microRNA expression profile in HepG2 cells treated for 24 h with atorvastatin, using a microarray platform. In the present study, we evaluated the expression of hsa-miR-17-5p, hsa-miR-20a-5p and hsa-miR-106a-5p in hypercholesterolemic patients before and after atorvastatin treatment and in HepG2 cells treated for 24 h with atorvastatin The miRNA hsa-mir-20a-5p was repressed after atorvastatin treatment in hypercholesteremic subjects and in HepG2 cells in culture. Repression of hsa-mir-20a-5p increased LDLR gene and protein expression in HepG2 cells, while hsa-mir-20a-5p overexpression reduced LDLR gene and protein expression.

## 1. Introduction

Cardiovascular diseases (CVD) are the leading cause of death in developed countries, being responsible for more than 17 million deaths annually. It is estimated that by 2030 this figure will rise to more than 23 million, thus confirming CVD will as the leading cause of death worldwide [1]. Ischemic heart disease and cerebrovascular disease are the most frequent CVD and have as their main common etiological mechanism the development of atherosclerosis [2]. One of the determining risk factors for atherosclerosis development is hypercholesterolemia, and the lipid-lowering therapy is based on the use of statins, which are the main therapeutic alternative for these patients [3,4,5]. Statins act by inhibiting the 3-hydroxy-3-methylglutaryl-coenzyme A (HMG-CoA) reductase, limiting the synthesis of cholesterol. This in turn promotes the activation of the transcription factors Sterol Regulatory Element-Binding Proteins (SREBPs) that induce the expression, among others, of the low-density lipoprotein receptor (*LDLR*) gene [6].

Despite the high rates of reduction in cholesterol levels reported after statin treatment, a wide interindividual variability in the response to statins treatment has been also described [7,8]. Numerous reports described genetic variants that can affect the pharmacokinetics and pharmacodynamics of statins, mainly referring to single-nucleotide polymorphisms (SNPs); however, these studies have not achieved the expected clinical impact. In addition, there is a percentage of patients (~10%) who develop intolerance to statin treatment, associated with muscle symptoms or high levels of creatine kinase [9,10,11,12,13]. These effects make it necessary to search for new lipid-lowering therapeutic alternatives.

MicroRNAs (miRNAs) are small non-coding RNAs (~22 nucleotides) involved in the post-transcriptional regulation of gene expression. Their expression is highly tissue- and cell-type specific. Due to their multi-target nature, they can be powerful tools to regulate pathological processes such as hypercholesterolemia and, therefore, to regulate plasma cholesterol levels in affected patients. This property makes them an attractive therapeutic alternative [14]. Among the miRNAs with therapeutic application, miR-122, miR-33-a and miR33-b stand out, being expressed specifically by the liver and located in the intron of the SREBP2 gene. In a study in non-human primates, inhibition of miR-122 by LNA-antimiR resulted in a 30% decrease in plasma cholesterol levels, particularly the low-density lipoprotein cholesterol (LDL-C) fraction, without any apparent liver toxicity [15]. Anti-miR-33 therapy on non-human primates showed an increase in plasma high-density lipoprotein cholesterol (HDL-C) levels and a decrease in very low density lipoprotein cholesterol (VLDL-C) [16].

Different studies showed the effect of lipid-lowering treatments with statins on the expression of miRNAs [17,18,19,20,21,22]. Recently, Ubilla et al. reported the overexpression of three microRNAs after atorvastatin treatment in hypercholesterolemic patients, describing the detection of microRNA miR-33b as a potential non-invasive biomarker of the response to atorvastatin [23]. Our group previously described the deregulation of 13 microRNAs in HepG2 cells treated with atorvastatin using a microarray platform; of these, the miRNAs hsa-miR-17-5p, hsa-miR-20a-5p and hsa-miR-106a-5p, which were decreased [22], would potentially bind to the 3′UTR region of the *LDLR* gene according to target prediction analysis.

In the present study, we evaluated the expression of the miRNAs hsa-miR-17-5p, hsa-miR-20a-5p and hsa-miR-106a-5p in hypercholesterolemic patients before and after atorvastatin treatment and the association between the expression of these miRNAs and the response to a lipid-lowering treatment with atorvastatin in hypercholesterolemic subjects. Secondly, we validated the expression of these miRNAs in HepG2 cells treated with atorvastatin and evaluated the effect of the modulation of deregulated miRNAs on the gene and protein expression of LDLR in HepG2 cells.

## 2. Results

### 2.1. Clinical and Demographic Characteristics

Table 1 shows the clinical and demographic characteristics of hypercholesterolemic individuals evaluated before statin treatment (baseline). None of the individuals experienced adverse effects.

The lipid profiles before and after treatment with 20 mg of atorvastatin/day/4 week are shown in Table 2. As it was expected, statin treatment reduced total cholesterol (T-CHO) and LDL-C levels, as well as the T-CHO/HDL-C index (*p* < 0.0001). However, there was no significant effect on HDL-C, VLDL-C and triglycerides (TG) levels (*p* > 0.05).

### 2.2. Atorvastatin Modulates the Expression of miRNAs hsa-miR-17-5p, hsa-miR-20a-5p and hsa-miR-106a-5p in Hypercholesterolemic Patients

The treatment with atorvastatin (20 mg/day/4 weeks) increased the expression of hsa-miR-17-5p (*p* = 0.0005) and decreased the expression of hsa-miR-20a-5p (*p* = 0.0006) and hsa-miR-106a-5p (*p* = 0.0004) in hypercholesterolemic patients (Figure 1).

A correlation analysis between miRNA expression and LDL-C reduction percentage did not show statistically significant results (Table 3).

### 2.3. Atorvastatin Decreases the Expression of miRNAs hsa-miR-17-5p and hsa-miR-20a-5p in HepG2 Cells Treated with Atorvastatin 10 μM

To validate the deregulate expression of hsa-miR-17-5p, hsa-miR-20a-5p and hsa-miR106a-5p determined using a microarray platform, the expression of miRNAs was evaluated by RT-qPCR in HepG2 cells treated with atorvastatin 10 μM for 24 h. These results showed a decreased expression of hsa-miR-17-5p (*p* < 0.0001) and hsa-miR-20a-5p (*p* = 0.0456) compared with vehicle treatment (Figure 2).

### 2.4. Modulation of miR-20a-5p Regulates LDLR Expression in HepG2 Cells

In silico analysis (TargetScan) predicted that hsa-miR-20a-5p binds to two conserved sites in the 3′UTR region of *LDLR* (Figure 3). To evaluate the effect of hsa-miR-20a-5p on *LDLR* expression, this miRNA was overexpressed or repressed by transfection of 10 nM, 30 nM and 60 nM mimics or inhibitors into HepG2 cells for 72 h. Overexpression of hsa-miR-20a-5p with 10 nM, 30 nM and 60 nM mimics significantly decreased the gene expression of *LDLR* (*p* < 0.0001) (Figure 4a). Similarly, LDLR protein expression was decreased using 60 nM hsa-miR-20a-5p mimic (*p* = 0.0203) (Figure 4c). The repression of hsa-miR-20a-5p with inhibitors at 10 nM, 30 nM and 60 nM concentrations significantly increased the gene expression of *LDLR* (*p* < 0.0001) (Figure 4b). Regarding protein expression, it increased when using miR-20a-5p inhibitor concentrations of 30 nM and 60 nM (*p* < 0.05) (Figure 4d).

## 3. Discussion

Statins are widely used worldwide in the treatment of hypercholesterolemia and may achieve a successful reduction up to 50% of the plasma levels of LDL-C, reducing the cardiovascular risk by up to 30% and improving the prognosis of patients with CVD [24,25,26]. However, a wide interindividual variability in response to statins treatment has been reported [7,8]. The study of genetic variations, mainly SNPs, has tried to explain this variability without being able to understand it in all cases [11]. This has led to important changes in statin treatment. The 2013 ACC/AHA (American College of Cardiology/American Heart Association) guidelines on blood cholesterol control emphasized the use of higher intensity statin therapies. In 2016, the ACC recommended the use of non-statin therapies, such as ezetimibe and PCSK9 inhibitors, and statin therapy at the maximum tolerated concentration in individuals whose LDL cholesterol and non-HDL cholesterol levels remain above certain thresholds after statin therapy [27].

The mechanism of action of statins is well known and is based on the inhibition of the limiting enzyme for cholesterol synthesis, HMG-CoA reductase. However, in recent years, the deregulation of miRNAs expression has also been described as a result of statin treatment [21,22,23,28]. Cerda et al. [21] recently described that atorvastatin mediated the upregulation of miR-129, miR-143, miR-205, miR-381 and miR-495 and the downregulation of miR-29b, and miR-33a in HepG2 cells. An in silico analysis of miRNA–mRNA interactions revealed a single network with six miRNAs modulating genes involved in lipogenesis and lipid metabolism [21]. Knowing the miRNAs profile changes in response to statin treatment and the role that these miRNAs would have in cholesterol regulation may be useful to understand other mechanisms of action of statins at the post-transcriptional regulatory level, identify biomarkers of response to treatment or identify new therapeutic targets that can regulate cholesterol metabolism, to be used as an alternative or complement to statin treatment.

Previously, our group evaluated the effect of statins treatment on the HepG2 cell miRNAoma using a microarray platform, where hsa-miR-17-5p, hsa-miR-20a-5p and hsa-miR-106a-5p, were downregulated as result of atorvastatin treatment [22]. Considering that target prediction analysis agreed that these three miRNAs could bind to the 3′UTR region of *LDLR*, a key gene in lipid metabolism responsible for the circulating LDL-C levels [6], in the present study we evaluated the expression of hsa-miR-17-5p, hsa-miR-20a-5p and hsa-miR-106a-5p in peripheral blood leukocytes of hypercholesterolemic patients before, before after treatment with atorvastatin. We observed a repressed expression of the miRNAs hsa-miR-20a-5p and hsa-miR-106a-5p, while miR-17-5p was overexpressed. Regarding the levels of these miRNAs in hypercholesterolemic patients, Ubilla et al. evaluated the expression of miR-17-5p in the plasma of hypercholesterolemic patients before and after treatment with atorvastatin, without observing significant differences [23]. This discrepancy can undoubtedly be due to the type of sample, since Ubilla et al. [23] used plasma, while we used leukocytes as a sample to measure the expression of miRNAs in hypercholesterolemic patients before and after treatment with atorvastatin. Among other reports, Xue et al. [29] described a significantly elevated expression of plasma miR-17-5p in patients with acute myocardial infarction, showing considerable diagnostic efficiency in this group of patients. There are no similar studies that associate the expression of hsa-miR-20a-5p and hsa-miR-106a-5p with the response to lipid-lowering treatment in patients.

Although the correlation analysis between the expression of the evaluated miRNAs and the LDL-C reduction percentage in hypercholesterolemic patients after atorvastatin treatment did not yield significant results, it would be interesting to evaluate the expression of these miRNAs in a larger cohort. This, with the purpose of identifying if there is a relationship between the expression levels of these miRNAs and the levels of response to atorvastatin treatment described by Zambrano et al., who reported that the miRNAs hsa-miR-106b-5p, hsa-miR-17-3p and hsa-miR-590-5p were found to be significantly repressed in peripheral blood mononuclear cells from hypercholesterolemic patients with a lower response to atorvastatin treatment [30]. In fact, it would be useful to identify response biomarkers to the treatment with atorvastatin or markers involved in the interindividual variability that has been widely described in the literature.

The validation of the expression of hsa-miR-17-5p, hsa-miR-20a-5p and hsa-miR-106a-5p in HepG2 cells treated with atorvastatin 10 μM for 24 h showed that the miRNAs hsa-miR-17-5p and has-miR-20a-5p of the cluster miR-17-92 were repressed. The cluster miR-17-92 is a polycistronic cluster that encodes six miRNAs, including miR-17-5p and miR-20a-5p. This cluster is highly conserved and related to the normal and pathological development of different organs, mainly associated with carcinogenic processes [31,32,33]. However, it is also associated with the normal and pathological cardiovascular development [34]. The miRNA hsa-miR-17-5p has been described to be significantly upregulated in hepatocellular carcinoma [35,36], and it has been reported that *LDLR* was repressed by overexpressing hsa-miR-17-5p in HeLa cells. The regulation of hsa-miR-17-5p on the 3′UTR region of *LDLR* was also demonstrated by a luciferase reporter assay [37]. Statins inhibit competitively and selectively the enzyme HMG-CoA reductase, limiting hepatic cholesterol synthesis. This promotes the activation of the transcription factors Sterol Regulatory Element-Binding Proteins (SREBPs) that induce the expression of the *LDLR* gene [38]. LDLR protein expression allows the uptake of circulating LDL, contributing to the lipid-lowering effects of statins. In silico analysis of miRNAs targets predicted that hsa-miR-20a-5p, like miR-17-5p, binds two conserved sites in the 3′UTR region of *LDLR*. There are numerous reports relating hsa-miR-20a-5p to carcinogenic processes, particularly to cell proliferation and migration [39,40,41]. However, there are no reports that relate this miRNA to cholesterol metabolism or studies validating the interaction of hsa-miR-20a-5p with *LDLR*. In our study, the miRNA hsa-mir-20a-5p was repressed after atorvastatin treatment in hypercholesteremic subjects and in HepG2 cells in culture. hsa-mir-20a-5p repression using inhibitors increased LDLR gene and protein expression in HepG2 cells, while hsa-mir-20a-5p overexpression using mimics reduced LDLR gene and protein expression. These data suggest that the miRNA hsa-mir-20a-5p could be involved in LDLR expression modulation. However, additional trials are needed to validate the interaction between miR-20a-5p and the 3′UTR region of *LDLR*.

This study has some limitations, such as the lack of validation of the interaction between hsa-miR-20a-5p and the 3′UTR of *LDLR* by a luciferase assay. Other studies could perform this validation and evaluate the impact of the miRNA hsa-mir-20a-5p on cholesterol metabolism through the modulation of LDLR. It would be interesting to evaluate if this possible modulation plays a role in the lipid-lowering mechanism of atorvastatin. Moreover, the small sample size implies that the results must be replicated in a wider population.

In conclusion, our results showed that the miRNA hsa-mir-20a-5p, which targets two sites in the 3′UTR region of the *LDLR*, is repressed by treatment with atorvastatin in hypercholesterolemic patients and in liver cells in vitro. Our results also suggest that hsa-miR-20a-5p should be one of the factors regulating LDLR expression. However, additional studies are necessary to validate this hypothesis.

## 4. Materials and Methods

### 4.1. Study Design and Participants

A total of 20 Chilean individuals diagnosed with hypercholesterolemia according to the NCEP criteria were selected for this study [35]. Patients were invited to the Chol-Chol Familial Health Center (Temuco, La Araucanía, Chile). Patients were treated with 20 mg/day of atorvastatin for four weeks. Individuals with familial hypercholesterolemia, hepatic or kidney disease, diabetes mellitus, endocrine disorders or malignant pathologies were excluded from the study. Additionally, patients taking medications such as diuretics, beta-blockers, concomitant lipid-lowering drugs, and drugs affecting their lipid profile, were excluded. The study protocol was approved by the Ethics Committee of Universidad de La Frontera (Protocol Nº084/19). All individuals voluntarily signed an informed consent.

### 4.2. Biochemical Analysis

To determine the concentration of serum lipids, blood samples were obtained before and after atorvastatin treatment by venous puncture after a 10–12 h overnight fast using vacutainer tubes without anticoagulant. Total cholesterol (T-CHO), triglycerides (TG) and high-density lipoproteins cholesterol (HDL-C) serum levels were determined using routine enzymatic colorimetric assays [42]. The low-density lipoproteins cholesterol (LDL-C) fraction was calculated using the Friedewald equation when triglycerides did not exceed 400 mg/dL [43].

### 4.3. RNA Extraction

EDTA-anticoagulated blood samples were obtained from hypercholesteremic patients before and after atorvastatin treatment. The buffy coat layer was separated by centrifugation and stored at −20 °C. Total RNA isolation and small RNA enrichment from buffy coat layer or HepG2 cells were performed using the *mir*Vana™ miRNA Isolation Kit (Invitrogen), according to the manufacturer’s instructions. RNA integrity analysis and quantification were performed using an Infinite^®^ 200 PRO NanoQuant (Tecan Group Ltd., Männedorf, Switzerland).

### 4.4. RT-qPCR

The relative expression of hsa-miR-17-5p, hsa-miR-20a-5p and hsa-miR-106a-5p in hypercholesteremic patients before and after atorvastatin treatment and in HepG2 cells treated with atorvastatin 10 µM were quantified by RT-qPCR. The cDNA templates were synthesized using the TaqMan™ Advanced miRNA cDNA Synthesis Kit (Applied Biosystems-Life Technologies, ThermoFisher, Waltham, MA, USA). Both procedures were realized according to the manufacturer’s protocol. Hsa-miR-17-5p, hsa-miR-20a-5p and hsa-miR-106a-5p expression was evaluated using TaqMan™ Advanced miRNA Assay (Applied Biosystems-Life Technologies, ThermoFisher, Waltham, MA, USA) and TaqMan^®^ Fast Advanced Master Mix (Applied Biosystems-Life Technologies, ThermoFisher, Waltham, MA, USA). PCR assays were performed in 48-well plates using a Step One Real-Time PCR system (Applied Biosystems-Life Technologies, ThermoFisher, Waltham, MA, USA). The thermal cycler protocol consisted of enzyme activation cycles at 95 °C for 20 s and 40 cycles at 95 °C for 1 s and 60 °C for 20 s. The 2^−ΔCt^ method [38] was used for data analysis and relative quantification of the amounts of transcripts in the sample. Endogenous control hsa-miR-191-5p was used as recommended by TaqMan™ Advanced miRNA Assay.

Relative mRNA expression of *LDLR* in HepG2 cells was determined in total RNA using TRIzol reagent (Invitrogen, Thermofisher, Waltham, MA, USA) in according to the manufacturer’s protocol. For mRNA quantification, a complementary DNA (cDNA) was synthesized using the High-Capacity cDNA Reverse Transcription Kit (Applied Biosystems-Life Technologies, ThermoFisher, Waltham, MA, USA). *LDLR* expression was evaluated using SYBR^®^ green master mix (Applied Biosystems-Life Technologies, ThermoFisher, Waltham, MA, USA). PCR assays were performed in 48-well plates using a Step One Real-Time PCR system (Applied Biosystems-Life Technologies, ThermoFisher, Waltham, MA, USA). The thermal cycler protocol consisted of enzyme activation cycles at 95 °C for 20 s and 40 cycles at 95 °C for 3 s and 60 °C for 30 s. All cDNA samples were assayed in triplicate. We tested target and reference genes on the same samples in separate tubes. The 2^−ΔΔCt^ method [44] was used for data analysis and relative quantification of the amount of transcripts in the sample. RPL27 was used as a reference gene, RPL13 and GAPDH were also evaluated in HepG2 cells using identical experimental conditions to determine the most stable and suitable reference gene according to the geNorm software (qBase+ 3.1 version, Biogazelle, Ghent, Belgium). The experiments were carried out in biological and technical triplicates.

### 4.5. Cell Culture and Reagents

Human HepG2 cells were cultured in Dulbecco’s modified Eagle medium (DMEM, Gibco™, ThermoFisher, Waltham, MA, USA) with 10% fetal bovine serum. The cells were cultured in the presence of 1% (*v*/*v*) penicillin/streptomycin at 37 °C in a humidified atmosphere containing 5% CO_2_. To evaluate the expression of hsa-miR-17-5p, hsa-miR-20a-5p and hsa-miR-106a-5p, HepG2 cells were seeded at 2.5 × 10^6^ cells per 75 cm^2^, cultured for 24 h, and treated with atorvastatin using concentrations of 0 μM (vehicle) and 10 μM for 24 h in 3 independent experiments [14]. To evaluate the effect of oligonucleotide transfection on LDLR expression, HepG2 cells were seeded at 5 × 10^4^ cells per 9.5 cm^2^ and cultured in the above-described conditions.

### 4.6. Oligonucleotide Transfection

To evaluate the effect of miRNAs on LDLR expression, transfection with mimics and inhibitors was performed as follows. All transfection experiments were conducted with Lipofectamine RNAiMAX Transfection Reagent (Invitrogen, ThermoFisher, Waltham, MA, USA), following the manufacturer’s instructions. Transfections were performed using *mir*Vana™ miRNA mimics and *mir*Vana™ miRNA Inhibitors (Life Technologies, ThermoFisher, Waltham, MA, USA). A mimic for hsa-miR-20a-5p (miRBase accession number MIMAT0000075, catalogue number 4464066) and an inhibitor for hsa-miR-20a-5p (miRBase accession number MIMAT0000075, catalogue number 4464084) were included. All experimental control conditions were treated with mirVana™ miRNA Mimic, Negative Control #1 (catalogue number 4464061) or mirVana™ miRNA Inhibitor, Negative Control #1 (catalog number 4464079). HepG2 cells were transfected with three concentrations of hsa-miR-20a-5p mimic (10 nM, 30 nM and 60 nM) or with three concentrations of hsa-miR-20a-5p inhibitor (10 nM, 30 nM and 60 nM) for 72 h. Both mimic and inhibitor transfection experiments were performed by using Opti-mem (Gibco™, ThermoFisher, Waltham, MA, USA) and Lipofectamine RNAiMAX (Invitrogen, ThermoFisher, Waltham, MA, USA) according to the manufacturer’s instructions.

### 4.7. Western Blot

HepG2 cells were directly lysed in RIPA buffer with proteases the Halt Protease & phosphatase Single-use Inhibitor cocktail (ThermoFisher, Waltham, MA, USA). After centrifugation for 5 min at 12,000× *g*, the supernatant was stored at −20 °C. Protein quantification was performed with the Protein Assay Dye Reagent (Bio-Rad, Hercules, CA, USA). Then, 40 μg of protein were denatured with Laemmli sample buffer (Bio-Rad, Hercules, CA, USA) during 5 min at 95 °C, loaded onto mini-PROTEAN TGX 4–20% of SDS-PAGE gels (Bio-Rad, Hercules, CA, USA) and transferred to PVDF membranes (Bio-Rad, Hercules, CA, USA). Primary antibodies anti-LDL receptor [EP1553Y] (1:1000, ab52818, Abcam, Cambridge, UK) and anti-beta actin [AC-15] (HRP) (1:20.000, ab49900, Abcam, Cambridge, UK) were used for protein detection. Chemiluminescence signals were detected using Supersignal West Pico Plus (ThermoFisher, Waltham, MA, USA) in a BOX Chemi XRQ (SYNGENE, Cambridge, UK) following the manufacturer’s instructions. Quantitative immunoblots were analyzed with ImageJ software (Version 1.46r, National Institutes of Health, Bethesda, MD, USA) [45]. LDLR levels were quantified and normalized using β-actin as an internal control.

### 4.8. Statistical Analysis

The results were analyzed using the statistic software GraphPad Prism version 5.0 (GraphPad Software Inc., San Diego, CA, USA). For demographic, clinical and laboratory variables, descriptive statistics was used. Continuous variables are shown as mean ± S.D. Sample size was estimated considering alpha and beta values of 0.05 and 0.2, respectively, and a size necessary to detect a four-fold difference between groups with at least 90% power resulting in 18 individuals in total. Values before and after treatment were compared using Wilcoxon matched pairs test after applying D’Agostino and Pearson omnibus normality test as distribution analysis. Comparisons between independent groups were performed with Student’s unpaired *t*-test. All cell culture experiments were performed independently three times. All values are two-tailed. Statistical significance was defined as *p* < 0.05.

## 5. Conclusions

In conclusion, our results showed that the miRNA hsa-mir-20a-5p, which targets two sites in the 3′UTR region of the *LDLR*, is repressed by treatment with atorvastatin in hypercholesterolemic patients and in liver cells in vitro. Our results also suggest that hsa-miR-20a-5p repression could be one of the mechanisms by which LDLR expression is regulated; however, additional studies are necessary to validate this hypothesis.

## Figures and Tables

**Figure 1 ijms-23-05022-f001:**
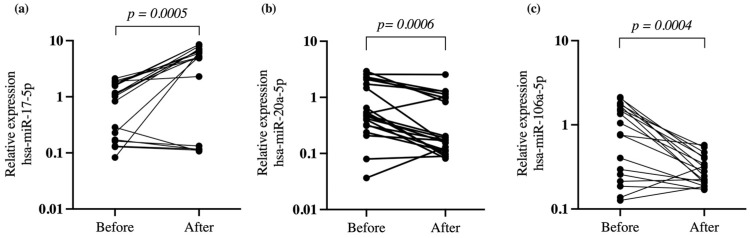
Relative expression of miRNAs in hypercholesteremic patients treated with atorvastatin. (**a**) hsa-miR-17-5p, (**b**) hsa-miR-20a-5p and (**c**) hsa-miR-106a-5p. The relative expression was determined by RT-qPCR from enriched RNA extracted from leukocyte cells in the peripheral blood of hypercholesteremic patients before and after atorvastatin treatment (20 mg/day/4 weeks). Normalization was performed using hsa-miR-191 as a reference gene; *p*-value was obtained by paired *t*-test.

**Figure 2 ijms-23-05022-f002:**
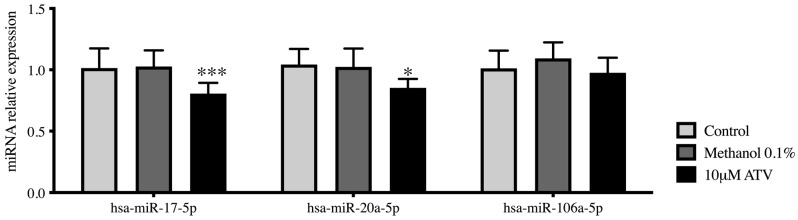
Relative expression of miRNAs in HepG2 cells treated with ATV 10 μM. Gene expression analysis of miRNAs hsa-miR-17-5p, hsa-miR-20a-5p and hsa-miR-106a-5p in control cells, vehicle-treated cells (methanol 0.1%) and cells treated with 10 μM ATV. * *p* = 0.0456 *** *p* < 0.0001 by unpaired *t*-test when compared with vehicle-treated cells (methanol 0.1%). The experiments were performed in technical and biological triplicates. ATV: atorvastatin.

**Figure 3 ijms-23-05022-f003:**
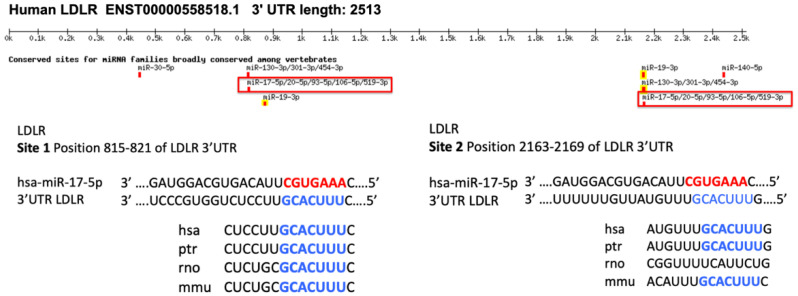
Binding sites for the miRNA hsa-miR-20a-5p in the 3′UTR region of *LDLR*. The 3′UTR region of *LDLR* in humans has two predicted and conserved binding sites for hsa-miR-20a-5p. The seed sequence of hsa-miR-20a-5p is shown in red, while its respective complementary region in the 3′UTR is shown in blue. Conserved sites between species are shown below: hsa, human; ptr, chimpanzee; mmu, mouse; rno, rat. LDLR: low-density lipoprotein receptor; UTR: untranslated region.

**Figure 4 ijms-23-05022-f004:**
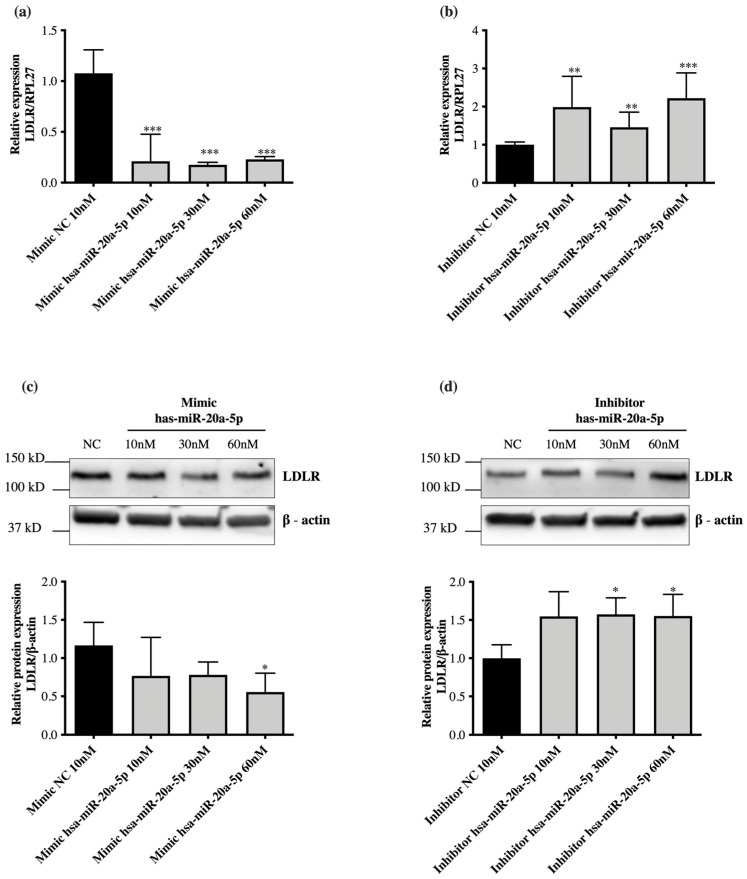
Post-transcriptional regulation of LDLR expression by hsa-miR-20a-5p in HepG2 cells. (**a**) *LDLR* gene expression analysis by RT-qPCR in HepG2 cells transfected with a negative control mimic (NC) or an hsa-miR-20a-5p mimic. (**b**) Gene expression analysis of *LDLR* by RT-qPCR in HepG2 cells transfected with a negative control inhibitor (NC) or an hsa-miR-20a-5p inhibitor. (**c**) LDLR protein expression analysis by western blot (representative image) of LDLR in HepG2 cells transfected with a negative control mimic (NC) or an hsa-miR-20a-5p mimics. (**d**) LDLR protein expression analysis by western blot (representative image) of LDLR in HepG2 cells transfected with a negative control inhibitor (NC) or an hsa-miR-20a-5p inhibitor. * *p* < 0.05; ** *p* < 0.005, *** *p* < 0.0001, compared to cells treated with negative control mimics or the inhibitor as appropriate, by unpaired *t*-test. The experiments were performed in technical and biological triplicates.

**Table 1 ijms-23-05022-t001:** Baseline characteristics of hypercholesteremic patients.

	Patients (*n* = 20)
Age (years)	47.3 ± 11.35
Sex (man/woman)	5/15
Hemoglobin (g/dL)	13.83 ± 1.15
Hematocrit (%)	41.25 ± 3.04
Leucocytes (×10^3^/µL)	6.68 ± 1.89
Platelets (×10^3^/µL)	243.6 ± 44.06
Glycemia (mg/dL)	95.4 ± 6.94
AST (U/L)	23.73 ± 5.33
ALT (U/L)	31.69 ± 8.59
CK (U/L)	110.2 ± 25.99
Blood creatin (mg/dL)	1.08 ± 0.16
Uremia (mg/dL)	30.88 ± 6.31
Ureic nitrogen (mg/dL)	13.31 ± 3.18
Total bilirubin (mg/dL)	0.63 ± 0.31
Direct bilirubin (mg/dL)	0.17 ± 0.13
Indirect bilirubin (mg/dL)	0.39 ± 0.25

Values are *n* or mean ± SD. AST, aspartate aminotransferase; ALT, alanine aminotransferase; CK, creatine kinase; SD, standard deviation.

**Table 2 ijms-23-05022-t002:** Lipid profile before and after a 4-week atorvastatin treatment in hypercholesterolemic patients.

	Before	After	*p*-Value
T-CHO (mg/dL)	239.4 ± 28.28	158.4 ± 33.41	<0.0001
HDL-C (mg/dL)	44.45 ± 10.09	41.20 ± 9.48	0.2127
LDL-C (mg/dL)	158.2 ± 33.41	91.37 ± 28.28	<0.0001
VLDL-C (mg/dL)	30.78 ± 13.04	24.19 ± 11.23	0.1611
TG (mg/dL)	150.4 ± 66.23	121.7 ± 55.40	0.2171
T-CHO/HDL-C	5.55 ±0.94	3.92 ± 0.80	<0.0001

Values are expressed as mean ± SD. T-CHO, total cholesterol; HDL-C, high-density lipoprotein cholesterol; LDL-C, low-density lipoprotein cholesterol; VLDL-C, very low-density lipoprotein cholesterol; TG, triglycerides; SD, standard deviation.

**Table 3 ijms-23-05022-t003:** Correlation analysis of miRNAs and the percentage of lipid reduction after atorvastatin treatment.

miRNA	Slope	*R* ^2^	*p*-Value
has-miR-17-5p	−0.3046 ± 0.2526	0.07473	0.2435
has-miR-20a-5p	−4.922 ± 5.388	0.04431	0.8345
has-miR-106a-5p	−9.939 ± 7.313	0.09308	0.1909

## Data Availability

The raw data supporting the conclusions of this article will be made available by the authors, without undue reservation, to any qualified researcher.

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
