# Peer review of "MicroRNA-20a-5p Downregulation by Atorvastatin: A Potential Mechanism Involved in Lipid-Lowering Therapy"

_ijms, 2022, doi:10.3390/ijms23095022_

Round 1

Reviewer 1 Report

The new potential mechanism of atorvastatin lipid-lowering therapy is discussed in the manuscript. The study concerns  the expression of hsa-miR-17-5p, hsa-miR-20a-5p and hsa-miR-106a-5p in hypercholesterolemic patients before and after atorvastatin treatment and in HepG2 cells treated for 24 hours with atorvastatin. In both experimental models hsa-miR-20a-5p was decreased. However, correlation analysis between miRNA expression and LDL reduction percentage did not show statistically significant results. Additional studies are necessary to confirm potential correlation between hsa-miR-20a-5p and LDLR. 

Some issues need clarification or improvement:

  1. hsa-miR-17-5p was up-regulated in patent-derived samples and down-regulated in HepG2 cells. Can you discuss the difference?
  2. Is the regulation of lipid metabolism the same in HepG2 cancer cells and in healthy hepatocytes?
  3. "The experiments were carried out in biological and technical triplicate", so what does mean: "All cDNA samples were assayed in duplicate."?
  4. line 162: " Regarding protein expression, it increased when using concentrations of 30nM miR-20a-5p inhibitor (p = 0.0030) (Figure 4d).", but Fig. 4d shows "*" also for 60nM.

Minor issues:

  1. Ad full name for ACC / AHA abbreviation.
  2. Abbrev. qRT-PCR and RT-qPCR were used; please unify.
  3. For suppliers in Materials and methods please add city, (state i USA), country.
  4. Line 345: 5x104 cells per ? cm2 or on which kind of culture plate?
  5. Line 359: weres --> were

Reviewer 2 Report

The presented draft by Saavedra et al discusses the potential role of MicroRNA-20a-5p in the lipid-lowering action of atorvastatin. 

The authors used human clinical samples (whole blood with further processing for mRNA isolation from the buffy coat), HepG2 cells plus standard biochemical techniques like western blotting, PCRs and cell culturing. 

I found the draft well-written, with the proper structure and correctly used American English vocabulary/grammar.  

I would like to express the major issues that formed during the peer-review process.

First of all, the Authors definitely overstated their conclusions and they should tone them down unquestionably. I do not think that the assessment of the role of micRNA might lead directly to explaining the exact mechanism of action of the atorvastatin. micRNA acts as a transponder/linker in the human body and for sure revealing the exact pathway/molecule up/down-regulated by the particular micRNA will show the true mechanism of the action.

Moreover, data presented for human samples is very restricted. Firstly, I am not sure how the Authors ended up in the conclusion that they need in total 18 samples for the study, as stated in their Statistical approach. Based on crucial clinical data and shown there estimations of SD and values the number should go up at least twice. Another problem is related to the lack of normo-cholesteric controls that will make your findings more reliable, since the values of lipid profile depend on many different factors. 

Another doubt came to my mind when analyzing the graph 1 (Figure 1 a and b). Please take into consideration that your population of the patient was very heterogeneous - their "dots" are split into two subgroups. It brings lots of variances as well as introduces potential bias. Please reconsider drawing this figure and estimate the effect of the treatment for a single patient. 

I am also not impressed by the fact that you used for in vitro studies HepG2 cells and for the "clinical" part of the study cells forming buffy coats. This is the very naive design that once again, introduces lots of potential bias.

The lack of the qualitative results in Table 3 might come from the heterogeneity of the studied population. In fact, it stands against your conclusions. 

At least one proinflammatory marker and one marker of the oxidative status should be evaluated in the patients' group.

I found also your very nicely written paper from 2021 published in Pharmacological Reports. You stated that you were analyzing the impact of i.e. atorvastatin in dosing of 0.1 - 10 uM on 84 types/forms of micRNA. Did you also evaluate in previously published paper micRNAs described here? 

Figure 3 is very blurry and hard to read (upper part).

The used dose is rather at the upper limit of using the studied drug in basic science. Did the authors check lower doses? Why particularly Atorvastatin has been chosen? 

What did lead the Authors to select these three particular micRNAs? 

Data presented on graph 4c and d is very weak. Should be interpreted carefully - very low drop (no more than 50%) as well as very mild increase (d). 

Please expand the limitation discussion - honestly, there are many other potential pitfalls. 

It would be also great to point out some further clinical applications of the presented data. 

Best. 

Round 2

Reviewer 1 Report

The manuscript was revised according to my suggestions.

Minor issue:

Please correct (4x) in section 4.6 spanish form of country to english form: Hercules, California, Estados Unidos  -->   shouled be Hercules, California, USA